# *Artemisia santolinifolia*-Mediated Chemosensitization via Activation of Distinct Cell Death Modes and Suppression of STAT3/Survivin-Signaling Pathways in NSCLC

**DOI:** 10.3390/molecules26237200

**Published:** 2021-11-27

**Authors:** Uyanga Batbold, Jun-Jen Liu

**Affiliations:** 1Ph.D. Program in Medical Biotechnology, College of Medical Science and Technology, Taipei Medical University, Taipei 11031, Taiwan; uyanga.gemini@yahoo.com; 2School of Medical Laboratory Science and Biotechnology, College of Medical Science and Technology, Taipei Medical University, Taipei 11031, Taiwan; 3Traditional Herbal Medicine Research Center, Taipei Medical University Hospital, Taipei 11031, Taiwan

**Keywords:** *Artemisia santolinifolia*, docetaxel, STAT3/survivin

## Abstract

Conventional chemotherapy remains an integral part of lung cancer therapy, regardless of its toxicity and drug resistance. Consequently, the discovery of an alternative to conventional chemotherapy is critical. *Artemisia santolinifolia* ethanol extract (AS) was assessed for its chemosensitizer ability when combined with the conventional anticancer drug, docetaxel (DTX), against non-small cell lung cancer (NSCLC). SRB assay was used to determine cell viability for A549 and H23 cell lines. The potential for this combination was examined by the combination index (CI). Further cell death, analyses with Annexin V/7AAD double staining, and corresponding protein expressions were analyzed. Surprisingly, AS synergistically enhanced the cytotoxic effect of DTX by inducing apoptosis in H23 cells through the caspase-dependent pathway, whereas selectively increased necrotic cell population in A549 cells, following the decline in GPX4 level and reactive oxygen species (ROS) activation with the highest rate in the combination treatment group. Furthermore, our results highlight the chemosensitization ability of AS when combined with DTX. It was closely associated with synergistic inhibition of oncogenesis signaling molecule STAT3 in both cell lines and concurrently downregulating prosurvival protein Survivin. Conclusively, AS could enhance DTX-induced cancer cells apoptosis by abrogating substantial prosurvival proteins’ expressions and triggering two distinct cell death pathways. Our data also highlight that AS might serve as an adjunctive therapeutic option along with a conventional chemotherapeutic agent in the management of NSCLC patients.

## 1. Introduction

It is estimated that more than two million new lung cancer cases are diagnosed in the world annually. Most of them are non-small cell lung cancer (NSCLC, 80%) [1]. Increasing knowledge of the underlying mechanisms and signaling pathways, particularly the invention of targeted therapy in NSCLC practice and outcomes from clinical trials, changed the chemotherapeutic strategy algorithm and shifted it to more personalized and selective treatment [2]. Although lung cancer incidence declines twice in men in prevalence of women, the 5-year relative survival rate is still low, as patients diagnosed in late stage [3]. Almost 77% of patients are not diagnosed until the disease has spread beyond the primary site. Thus, chemotherapy remains the foundation form of their treatment, regardless of its toxicity and resistance [4,5,6].

Traditional medicine has been used since ancient times and has gained increasing attention as a remarkable source for finding new drugs with few side effects [7]. The benefits of combinational treatment of traditional medicine with conventional chemotherapy are not only restricted by synergistically boosting treatment performance, minimizing the evidence of drug resistance, and also exerting chemoprotective actions [8]. Nowadays, scientists believe that identifying the bioactive components is vital to modernizing herbal extracts for clinical use. However, there will be a loss of synergetic action of the pharmaceutical ingredients without the appropriate approaches. Exploring the mechanism of synergistic action for combining herbal extracts with conventional chemo drugs will help researchers discover both new phytomedicines and new drug combinations [9]. Consequently, combinations of natural products with chemo drugs have been used in several clinical trials against various malignancies, including lung cancer (especially non-small-cell lung cancer) since 2006. The chemotherapeutic agents used for these trials are, in most cases, platinum-based drugs, such as cisplatin and oxaliplatin, in addition to docetaxel, even though these drugs could cause severe side effects [10].

Docetaxel (DTX), an antimicrotubule taxane binding to the β-tubulin subunit of microtubulin, is a standard regimen for both first- and second-line treatment in advanced NSCLC [11,12]. Although survival rates after docetaxel monotherapy in chemotherapy-naive patients were similar to those platinum-based standard regimens in NSCLC patients, docetaxel is limited in use by its severe side effects and drug resistance [13].

*Artemisia santolinifolia* is a “wormwood” semi-shrub with three-pinnatisect leaves, widely distributed in Central Asia to Eastern Europe, ranging between 400 and 500 different species [14]. *A. santolinifolia* belongs to the large and diverse genus of Artemisia, that have been introduced as a common widespread species. However, little is known about its biological activity [15]. The chemical constituents of *A. santolinifolia*, such as terpenoids, polyacetylenes, coumarins, glycosides, sterols, and flavonoids, were documented in several studies [16,17,18]. Furthermore, previous studies have reported in vitro and in vivo cytotoxic and antitumor effects of various Artemisia species and the related molecular pathways, such as activation of caspase, modulation of Bax/Bcl-2 ratio and generation of ROS level, subsequent cell cycle arrest at S, G2/M phase, as well as inhibition of specific target molecules—notch 1, β-catenin, and MMP9 [19,20,21]. However, the anticancer activity of *A. santolinifolia* combined with conventional chemotherapeutic docetaxel (DTX) has not yet been investigated.

In this study, we utilized two NSCLC cell lines (A549 and H23) to explore the synergizing effect and the possible mechanism of action of AS when combined with conventional chemotherapeutic agent DTX. A remarkable finding in this study was that the expression of survivin in both cell lines was markedly reduced by STAT3 inhibition, despite showing different patterns of cell death in the flow cytometry analysis. Additionally, AS showed the selective mode of enhancement of DTX-induced cell death. In A549 cells, AS synergistically decreased cell viability, depleted GPX4 protein levels, and increased ROS generation, suggesting the predominant contribution of ferroptosis. In contrast, AS increases DTX-responsive apoptosis-related proteins in the H23 cell line as a different mode of cell death. Our study demonstrated that AS can be a promising chemosensitizer with the combination of conventional chemotherapeutic agent DTX for NSCLC.

## 2. Results

### 2.1. AS Combined with DTX Synergistically Inhibits A549 and H23 Cells Proliferation

Using natural product-based alternative treatment approaches and conventional cytotoxic drugs could facilitate chemosensitization in cancer cells, primarily due to biological activity in a multitargeted manner, thereby influencing multiple regulatory pathways to improve the overall chemotherapeutic response in various cancers [22]. Therefore, we hypothesized that co-treatment of AS with DTX would allow a lower dose of anticancer drug to minimize undesired harmful side effects. Firstly, A549 and H23 cells were treated with series doses of AS and DTX solely to determine an initial dose–response. The obtained data indicated that IC_50_ of DTX and AS for A549 were 3.6 nM and 238 µg/mL, while for H23, they were 3.8 nM and 266.7 µg/mL after 48 h (Figure 1A,B). Further, to analyze whether the combination of AS and DTX could possess a synergistic anticancer effect, cells were concurrently treated with AS at concentrations of 100 and 200 μg/mL in combination with DTX at concentrations of 0.5, 1 and 2.5 nM for 48 h. The combination indexes (CIs) were calculated using CompuSyn software (Table 1). In general, synergism (CI < 1) was observed at high inhibition levels in concurrent treatment with a low dose of DTX, while additivity to antagonism (CI = 1, CI > 1) was observed at the high dose of DTX. Among the three regimens, co-treatment groups in both cell lines with AS and chemo drug markedly enhanced the growth inhibitory effect of 0.5 nM DTX (CI 0.68 for A549, CI 0.78 for H23), compared with that of 2.5 nM DTX alone. Furthermore, especially for A549 cells, combination regimens of DTX at concentration 0.5 nM and AS co-treatment resulted in more significant cytotoxicity effects with statistical significance than the individual agents alone after 24 h, 48 h, 72 h (*p* < 0.01, Figure 1C). Hence, for further experiments, we selected a low dose of DTX (0.5 nM), which reduction in cell proliferation is limited to 20%, and a fixed concentration of AS (100 and 200 µg/mL) to investigate the mechanism of synergistic action.

### 2.2. Enhancement of DTX-Induced Cytotoxicity Effect through Distinct Cell Death Modalities

To study the synergistic effect of AS and DTX, Annexin V/7-amino-actinomycin D (7-AAD) double staining was performed on cell cytotoxicity toward A549, H23 cell lines. Our results showed co-administration of AS and DTX has an apparent synergistic effect after 48 h on both cell lines in the induction of apoptosis with statistical significance, especially in H23 cells (*p* < 0.05, Figure 2C). However, the results showed a higher necrotic population (AnnexinV^−^/7-AAD^+^), specifically in the A549 cell line, when treated with AS alone and in the combined treatment group (*p* < 0.05, Figure 2A). As shown in Figure 2B, AS, alone or merged with DTX, induced less necrosis but more apoptotic death in the H23 cell line.

### 2.3. AS-Mediated Chemosensitization in H23 Is Primarily through Activation of Apoptosis, While in A549 via Ferroptosis

In our previous study, we observed that AS alone could inhibit NSCLC growth through distinct cell death modes depending on the cell line, with concomitant generation of cellular ROS level, induction lipid peroxidation, and reduction in glutathione peroxidase 4 (GPX4) expression selectively in A549, while having triggered caspase-dependent apoptosis in H23 cells. Therefore, in the present study, we proposed to uncover the role of relative protein expressions in the mechanism by which AS enhances the cytotoxicity while co-treated with DTX. To this end, we first analyzed corresponding protein expressions in both cell lines, which could be upregulated in response to exposure of DTX (0.5 nM) alone or combined with AS (100 and 200 µg/mL) for 48 h. As shown in Figure 3A,B, co-treatment groups showed more decrease in procaspase 3 protein in a concentration-dependent manner and a higher increase in cleaved caspase 3 protein than those in either AS or DTX groups alone only in H23 cells, but no significant alterations were detected in A549 cells in all treatment groups (*p* < 0.01, Figure 3A,B). These results suggested that AS could enhance DTX-induced apoptosis of NSCLC cells by primarily triggering caspase-dependent apoptosis in H23 cells. Conversely, further validation of ferroptosis-related specific marker GPX4 expression showed remarkable inhibition with statistical significance in A549 cells (*p* < 0.001, Figure 3C,D). At the same time, there were no aberrant changes either in procaspase 3 or cleaved caspase 3 expressions in A549, highlighting the specific effects of AS in mediating chemosensitization through ferroptosis in A549. The degree of ferroptosis was evaluated by testing the ROS levels. ROS levels were significantly higher in A549 cells (*p* < 0.001, Figure 3E,F), either when treated with AS individually or along with DTX, but no apparent alterations were detected in H23. Nevertheless, the combination treatment group demonstrated a higher generation of ROS level following treatment with AS plus DTX versus AS alone, suggesting ROS expression is among the potential vital mechanisms contributing to the synergizing effect of AS through ferroptosis selectively in A549.

### 2.4. The Enhancement Functioning of AS on DTX-Induced Cell Death Associated with Inhibition of STAT3/Survivin Signaling

To elucidate whether the additive effect of AS on DTX-induced apoptosis was mediated by STAT3 signal transducer and downstream prosurvival protein, we evaluated the expression levels of STAT3, p-STAT3, and survivin in NSCLC cells after 48 h of drug exposure. AS and DTX solely inhibited STAT3 phosphorylation, and many more in the combined group in both cell lines with rates of approximately 80.8 ± 0.2% in A549 cells (*p* < 0.001) and 78.7 ± 1.7% in H23 cells (*p* < 0.05) at 48 h, respectively, compared with the internal control β-actin (Figure 4A,B). Among all three treatment groups, the most inhibition effect downstream of the STAT3 signaling molecule, survivin, were quantified in combined treatment groups with statistical significance and with relative rates of inhibition of 56.3 ± 0.2 in A549 (*p* < 0.001) and 37.6 ± 1.0% in H23, respectively (Figure 4C). It was intriguing to observe that survivin expression was significantly downregulated by AS individually and in combination groups in both cell lines, whereas DTX treatment by itself did not alter its expression, highlighting the specific effects of AS in mediating chemosensitization. According to the results, either phosphorylation of STAT3 or prosurvival protein survivin expression could be more downregulated by combined treatment groups in both cell lines, suggesting the function of AS enhancing DTX-induced antitumor activity through suppression of prosurvival proteins in NSCLC.

### 2.5. LC-QTOF Analysis of Ethanol Extract of A. santolinifolia

To ascertain the possible chemical composition–function relationship, AS was analyzed using LC-QTOF. The analysis of ethanol extract by LC-QTOF showed that the non-polar chemical constituents of *A. santolinifolia* contained more than one class of natural product compounds, such as steroids, terpenoids, monoterpenes, phenolic compounds, and glycolipids. Accordingly, Table 2 summarizes the eight most abundant compounds identified in AS with their list of reference sources that reported potential anticancer effects.

## 3. Discussion

Combination cancer therapies reduce side effects, combat drug resistance, and improve synergy, allowing the effectiveness of the combined therapeutics agents to surpass the results of each on their own [31]. The multi-component structure of medicinal herbs makes them particularly suitable for treating complex diseases such as cancer. However, the mechanism of action of the whole extracts of medicinal herbs remains largely unclear [32]. In this study, for the first time, we aimed to investigate AS contribution toward enhancement effect of DTX-induced cell death and elucidate the possible mechanism of action. AS merged with DTX significantly inhibited the viability of the cell, compared with either agent alone in A549 and H23 cell lines. In combination therapy with a lower DTX dose (0.5 nM) than the regular regime, the CI showed greater synergy in both cells after 48 h treatment. Interestingly, when administered alone or in combination, AS induced an entry point marker in the apoptotic signaling pathway caspase-3 in H23 cells but not in A549. This finding suggests that AS potentiated chemosensitization through activation of apoptosis.

Research studies have demonstrated the efficacy of inducing newly discovered cancer cell death through ferroptosis as a therapeutic approach for chemotherapy-resistant cancer cells [33]. Ferroptosis may also affect the effectiveness of chemotherapy, radiotherapy, and immunotherapy. Therefore, combinations of agents that target ferroptosis signaling could improve the outcome of those therapies [34]. Previously, we have identified that AS could alter the ROS generation and expression of specific ferroptosis marker protein GPX4 selectively in the A549 cell lines. In agreement with these results, despite no evidence of synergistic cleavage of caspase-3 protein upon combination treatment in A549, the accumulation of ROS level was significantly higher than the individual agents alone. Moreover, AS merged with DTX significantly inhibited GPX4 expression, but no noticeable alteration was detected in the H23 cells. Collectively, we suspect that ferroptosis may play a predominant role in AS enhancement of cell death combined treatment with DTX in A549 cells selectively, whereas H23 cells’ chemosensitization is primarily caused via apoptosis (Figure 5).

It has been reported that signal transducer and activator of transcription 3 (STAT3) was activated in 22–65% of NSCLC [35]. Targeting STAT3, either directly or by inhibiting upstream regulators, further impairs their survival, contributing to oncogenesis by either preventing apoptosis or enhancing cell proliferation [36]. STAT3 previously has been reported to modulate the expression of genes involved in cell apoptosis and ferroptosis [32]. Thus, we speculated that STAT3-induced inhibition of apoptosis may be involved in the survival of cancer cells after chemotherapeutic agent exposure in NSCLC. Furthermore, STAT3 is described as potentially regulating genes that may confer resistance to apoptosis. Therefore, the expression of signal molecular downstream of STAT3, such as survivin, should be monitored [37]. Survivin is the smallest member of the inhibitor of apoptosis (IAP) family, and its expression is almost undetectable in normal adult tissues but found to be overexpressed in most human malignancies [38]. A natural compound, the indole-based tambjamine analog 21 (T21), was previously reported to kill cancer cells through inhibiting survivin protein expression in vitro, and the molecular mechanism involved in cell death was by blocking of Janus kinase/signal transducer and activator of transcription-3 (JAK/STAT3)/survivin pathway [39]. Thus, we reasoned that STAT3/survivin signaling might reveal how AS sensitizes or enhances the effect of DTX in lung cancer cells. Subsequent Western blot analysis revealed that AS solely or co-treated with DTX could strongly inhibit the expression of p-STAT3 and downstream molecular target survivin in both A549, H23 cells, regardless of a distinct mode of cell death. AS inhibited STAT3 expression, and the combination of AS with DTX had the most inhibition effect on the phosphorylation of STAT3.

The chemical compounds identified by LC-QTOF analysis hits were mostly steroids and terpenoids. In this report, we highlight the two major compounds that are more likely to possess the potential ability of chemosensitization of NSCLC. The principal phytosterol constituent of riceberry rice bran (RBDS), gramisterol, which is consistent with our results, was studied for cytotoxicity in WEHI-3 cells. Gramisterol potentiated a good growth-inhibitory effect on leukemic cells through abrogation of p-STAT3 signaling [23]. Trametenolic acid B (TAB), which belongs to the class of triterpenoids, demonstrated cytotoxic effect on HGC-27 cells but had no apparent signs of cell apoptosis. Instead, TAB induced autophagic cell death in gastric cancer cells [40]. Recent evidence showed that TAB could sensitize breast cancer MDA-MB-231/Taxol cells to Taxol by increasing the intracellular accumulation of the chemo drug via abrogating the expression of major drug efflux P-gp protein [27].

Collectively, our results indicate that suppression of STAT3/survivin signaling pathway and selective induction of ROS generation is among the fundamental mechanisms contributing to the particular chemosensitizing properties of AS in NSCLC. Additionally, our study highlights the medicinal herb AS as a discerning ferroptosis and apoptosis inducer.

## 4. Materials and Methods

### 4.1. Collection of Plant Material

*Artemisia santolinifolia* was obtained from a Mongolian traditional herb medicine company (Mong-Em; Ulaanbaatar, Mongolia), collected in August 2020. Herb was air-dried for further extraction process.

### 4.2. Preparation of Extract

*Artemisia santolinifolia* ethanol extract (AS) was prepared as follows: The dried plant material was soaked in 99% ethanol (1:1 *w*/*v*) and slowly stirred at 60 °C for 6 h. After that, the supernatant was filtered through filter paper, and the solid material was removed. The ethanol extract was concentrated using a Rotavapor (Buchi Labortechnik; Flawil, Switzerland) to remove solvent at 40 °C and reconstituted in 99% ETOH at various concentrations for in vitro studies.

### 4.3. Cell Culture

The human adenocarcinoma cell line A549 and NCI-H23 were obtained from ATCC Cell Bank and cultured in DMEM/F12, RPMI 1640 (Gibco, Carlsbad, CA, USA) with 10% heat-inactivated FBS (Fetal Bovine Serum, Biological Industries, Cromwell, CT, USA) in the presence of 1% L-glutamine (Corning, NY, USA) and 1% penicillin–streptomycin solution, 100 U/mL (Corning, NY, USA) at 37 °C in a humidified atmosphere containing 5% CO_2_ before use in experiments.

### 4.4. SRB Assay

Cell viability was examined as previously described [41]. Briefly, cells were seeded on 96-well plates at 2.5 × 10^3^/well of A549 and 3 × 10^3^/well of H23 overnight. The cells were treated with different concentrations of AS or/and DTX for 48 h. Afterward, cells were washed twice with 200 μL phosphate-buffered saline (PBS) and then fixed in trichloroacetic acid (TCA, Sigma-Aldrich; St Louis, MO, USA) for 1 h. Cells were stained with sulforhodamine B (SRB, Sigma Aldrich; St Louis, MO, USA) and then washed with a 1% acetic acid solution three times. The plate was air-dried and solubilized with a Tris base solution (BioShop; Ontario, Canada). The absorbance change of each well was assessed at the wavelength of 515 nm using the SpektraMax iD3 multi-mode microplate reader (Molecular Devices; CA, United States).

### 4.5. Drug Synergy

The inhibition rate of cell viability was detected, while the different doses of AS (100 and 200 μg/mL) and DTX (0.5, 2.5 and 5 nM) were added solely or combined for a period of 48 h to evaluate the feasibility of AS and DTX. The combination index (CI) and the combinatorial effect of AS and DTX were investigated by the Chou–Talalay equation [42]. A synergistic effect occurs when the CI value is <1; additive occurs when the CI value equals 1, and antagonism when the CI value is >1.

### 4.6. Cell Cycle Analysis

Cell cycle progression was analyzed by flow cytometry. A549 and H23 cells were seeded in 60 mm culture dishes at a density of 3 × 10^5^/5 mL and cultured overnight. Subsequently, cells were treated with increasing concentrations of AS (100 and 200 µg/mL) alone or combined with DTX (0.5 nM) for 48 h. After treatment, the cells were harvested by trypsinization, followed by centrifugation at 2000 rpm for 5 min. The cells were fixed in ice-cold 70% ethanol at −20 °C for 24 h. Prior to flow cytometric analysis, the ethanol was removed by centrifugation. After washing with PBS, 3 μL of propidium iodide (PI, 1 mg/mL) stain solution and 5 μL RNase A (1 mg/mL) (Sigma-Aldrich; St Louis, MO, USA) were added to samples and placed for 30 min at room temperature. Cell cycle analysis was performed using an Attune NxT flow cytometer (Invitrogen; Carlsbad, CA, USA) at an excitation wavelength of 488 nm and an emission wavelength of 610 nm by measuring the amount of PI-labeled DNA in the cells. Data were acquired from 10,000 cells and were analyzed using FlowJo software (TreeStar; Ashland, OR, USA).

### 4.7. Flow Cytometric Analysis of Early and Late Apoptosis

Flow cytometric analysis was performed using FITC Annexin V Apoptosis Detection Kit (Biolegend; San Diego, CA, USA). A549 and H23 cells were harvested via trypsin-EDTA treatment, washed twice with PBS, and centrifuged. The pellets were resuspended and stained with 5 µL FITC-conjugated Annexin V for 10 min and then 5 µL 7-AAD (7-amino-actinomycin D) for 5 min at room temperature in the dark. For the analysis, 500 µL binding buffer was added to each mixture and analyzed using Attune NxT flow cytometer (Invitrogen; Carlsbad, CA, USA). The results were analyzed using FlowJo (TreeStar; Ashland, OR, USA) software. The ratio of apoptotic cells was calculated by the percentage of each of early and late apoptotic cells.

### 4.8. Analysis of ROS

Intracellular ROS was evaluated by using 2,7′-dichlorodihydrofluorescein diacetate (H_2_DCFH-DA) kit from Sigma-Aldrich (St. Louis, MO, USA). After the above-indicated treatments for 48 h, cells were stained with 5 µM DCFH-DA for 30 min in an incubator and then washed twice with phosphate-buffered saline (PBS). The fluorescence intensity of cells was analyzed by flow cytometry. Data analysis was performed using FlowJo software (TreeStar; Ashland, OR, USA).

### 4.9. Western Blot Analysis

Briefly, the cells were treated with AS (100, 200 µg/mL) or/and DTX (0.5 nM) for 48 h. Cellular lysates were centrifuged at 12,500 rpm for 30 min, and the supernatant was collected. Thirty micrograms of protein samples were separated by SDS-PAGE and then transferred onto a nylon membrane for Western blot analysis. The membranes were blocked with TBS-5% non-fat milk for 1–2 hrs at room temperature and subsequently probed with primary antibodies: against caspase-3, survivin (Cell Signaling Technology; Danvers, MA, USA), GPX4 (Gentex; Irvine, CA, USA), and STAT3 (Abcam; Cambridge, UK) overnight at 4 °C, diluted according to the protocol. After washing with TBS-T, the blots were incubated with appropriate HRP-conjugated secondary antibody for 1 h at room temperature. Protein bands were detected by using ECL chemiluminescence and exposed to X-ray film ImageQuant (LAS 4000 series) for 1–60 s. β-actin (Abcam; Cambridge, UK) was used as the internal control.

### 4.10. LC-MS/MS Analysis

The analytical LC-MS/MS experiment was performed on a Waters ACQUITY UPLC I-Class system and Vion IMF QTof MS spectrophotometry. The system is equipped with a Waters BEH C18 Acquity analytical column (75 μm × 150 mm, 1.8 μm). The column oven temperature was set at 40 °C, and the temperature of the auto-sampler was set at 4 °C. For each LC-HDMSE run, approximately 2 μL of the sample (1 mg/mL) was loaded onto the column through a 10 μL sample-loop using 98% mobile phase A (0.1% formic acid in H2O) at a flow rate of 0.4 mL/min with a gradient elution consisting of an increase from 20% to 46% mobile phase B (0.1% formic acid in ACN) over 30 min, and a re-equilibration step at 20% mobile phase B for 10 min. The lock mass, 200 fmol/μL of [Leu] solution prepared with 0.1% formic acid in 30% ACN, was delivered from the auxiliary pump at a flow rate of 0.2 μL/min to the reference sprayer of the LockSpray source.

LC-HDMSE Data were acquired in resolution mode with UNIFI Scientific Information System. The mass spectrometer operated in resolution mode with a typical resolving power of at least 40,000 FWHM at *m*/*z* 500. All analyses were performed using positive mode ESI using a LockSpray source. The lock mass channel was sampled every 30 s. The mass spectrometer was calibrated with a Leucine enkephalin solution (200 fmol/μL) delivered through the reference sprayer of the LockSpray source. Accurate mass LC-HDMSE data were acquired in an alternating, low energy (MS) and high energy (MSE) mode of acquisition with a mass scan range from *m*/*z* 50 to 1000, using a capillary voltage of 2 kV, a source temperature of 150 °C, and a cone voltage of 30 V. The mass measured accuracy was <5 ppm when the instrument operated with the lock mass, and the accuracy was the same between low energy and high energy MS scans. The spectral acquisition time in each mode was 1.0 s with a 0.1 s inter-scan delay. In low-energy HDMS mode, data were collected at a constant collision energy of 2 eV in the trap and transfer cells. In high-energy HDMSE mode, the collision energy was ramped up from 15 to 30 eV in the transfer cell only.

### 4.11. Statistics

All data were treated with Sigma Plot (Systat Software; San Jose, CA, USA) software. The results are presented as means ±SE of three independent experiments. Differences between the two groups were calculated using Student’s *t*-test and analysis of variance (ANOVA) with pair-wise comparisons. Statistical significance was defined at * *p* < 0.05, ** *p* < 0.01, and *** *p* < 0.001 vs. corresponding control.

## Figures and Tables

**Figure 1 molecules-26-07200-f001:**
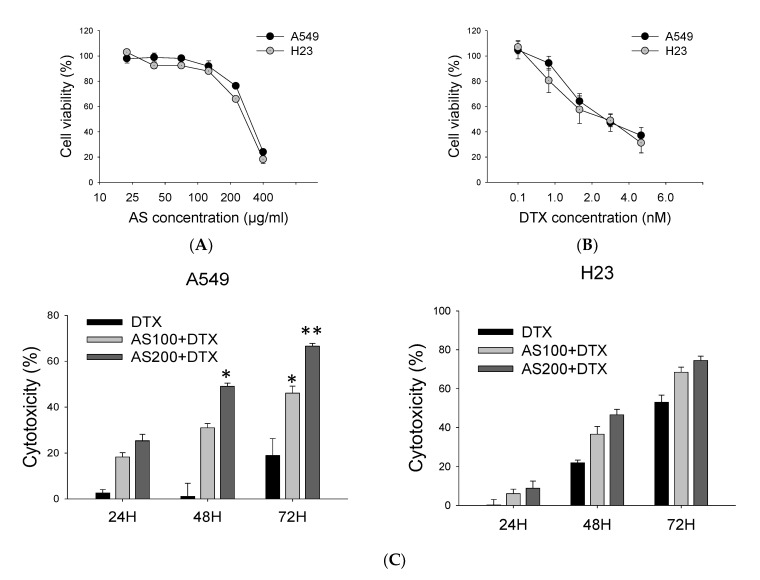
Effect of AS and/or DTX on NSCLC cells viability. Cell viability was assessed by SRB assay. Preliminary dose–response study on A549 and H23 cells treated with series of AS concentrations (**A**) and DTX for 48 h (**B**). (**C**) Inhibitory effects of AS (100, 200 μg/mL) combined with a sub-optimal dose of DTX (0.5 nM) on the proliferation of A549, H23 cells for 24 h, 48 h and 72 h. Data are the mean ±SE; * *p* < 0.05, ** *p* < 0.01 vs. corresponding control.

**Figure 2 molecules-26-07200-f002:**
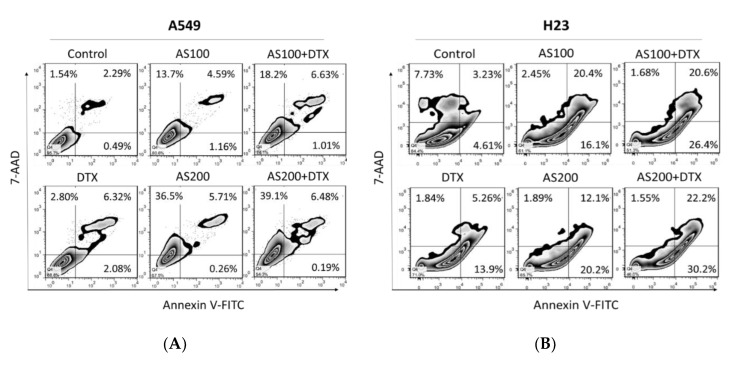
AS enhances DTX-induced apoptosis in NSCLC cells inducing different features of cell death. Analysis of cell death in NSCLC cells following AS and/or DTX treatment was performed through FCM with Annexin V/7-AAD staining. A549 (**A**) and H23 (**B**) cells were treated with 100 and 200 µg/mL AS and 0.5 nM DTX alone or in combination for 48 h. (**C**) The representative histograms and quantified results of three independent experiments are shown. Data are the mean ± SE; * *p* < 0.05 vs. corresponding control.

**Figure 3 molecules-26-07200-f003:**
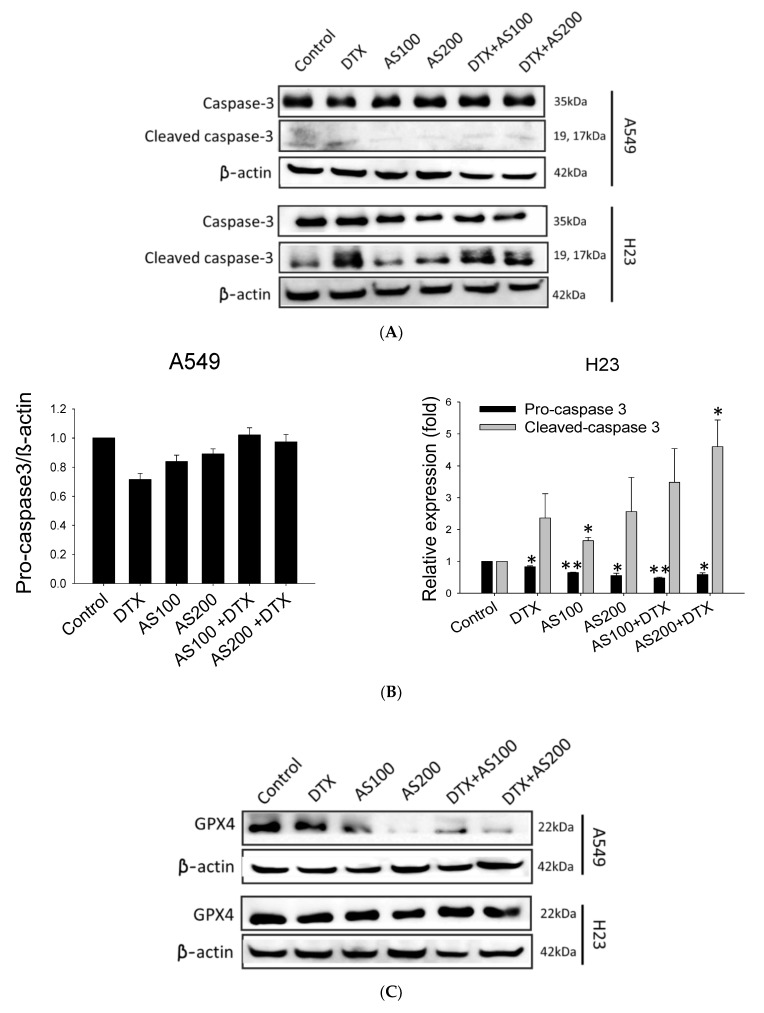
Apoptosis and ferroptosis contributed to AS-induced chemosensitization along with DTX in lung cancer cells: (**A**) the cells were treated with AS (100 and 200 μg/mL) alone and/or in combination with DTX (0.5 nM) for 48 h. Western blot results of proteins with the most significant alterations of pro-caspase 3, cleaved caspase 3 levels in A549 and H23 cells were validated; (**B**) the representative histograms quantified results of three independent experiments are shown; (**C**) the expression of key ferroptosis regulators GPX4 was examined by Western blotting following treatment as indicated above; (**D**) representative histograms of corresponding quantitative analyses of GPX4; (**E**,**F**) A flow cytometer analyzed the cellular ROS level in A549 and H23 cells with representative histograms. Data are the mean ±SE; * *p* < 0.05, ** *p* < 0.01, *** *p <* 0.001 vs. corresponding control.

**Figure 4 molecules-26-07200-f004:**
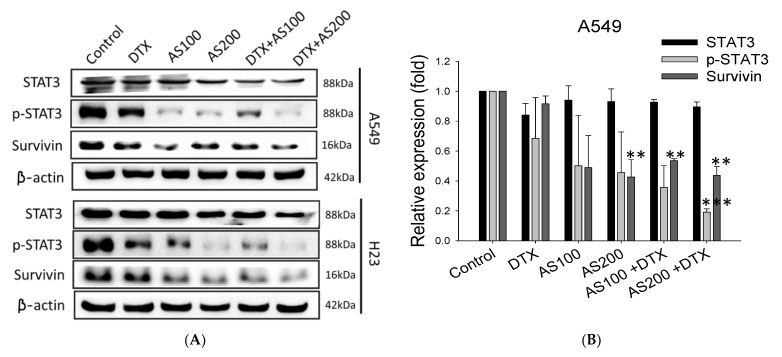
Synergizing effect of AS and DTX on the expressions of STAT3/survivin signaling. A549 and H23 cells were treated with AS (100, 200 μg/mL) and DTX (0.5 nM) alone or the two-drug combination for 48 h: (**A**) STAT3, p-STAT3, and survivin proteins levels were investigated. Western immunoblotting was performed and β-actin was used as loading control; (**B**,**C**) representative histograms of corresponding quantitative analyses. Data are the mean ±SE; * *p <* 0.05, ** *p* < 0.01, *** *p* < 0.001 vs. corresponding control.

**Figure 5 molecules-26-07200-f005:**
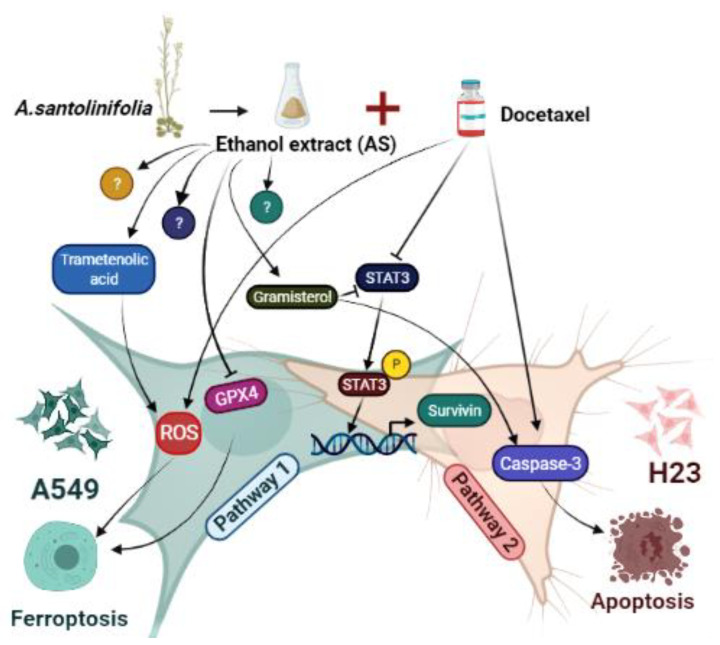
AS-mediated chemosensitization via activation of distinct cell death modes in NSCLC. Black arrows indicate the activity of AS and DTX; blocked black arrows denote inhibited signaling molecules.

**Table 1 molecules-26-07200-t001:** Combination index (CI) of concurrent treatment of AS and DTX after 48, whereas CI < 1 indicated synergistic effect; CI = 1 additivity, while CI > 1 indicated antagonism.

Cell Line	AS (µg/mL)	DTX 0.0 nM	DTX 0.5 nM	DTX 1 nM	DTX 2.5 nM
**A549**	0.0		8.8 ± 6.5	23.7 ± 2.7	47.1 ± 1.9
	100	18.4 ± 3.4	30.9 ± 1.9 (0.96)	40.2 ± 1.1 (0.97)	45.7 ± 3.5(1.09)
	200	37.3 ± 1.4	58.7 ± 3.5 (0.68)	59.3 ± 1.5 (0.81)	65.4 ± 0.5 (1.04)
**H23**	0.0		20.8 ± 4.4	36.1 ± 5.7	42.2 ± 5.8
	100	14.8 ± 3.3	37.2 ± 3.0 (0.74)	42.1 ± 1.7 (0.82)	49.3 ± 4.4 (1.01)
	200	31.2 ± 2.7	47.2 ± 2.8 (0.78)	53.1 ± 2.2 (0.74)	54.4 ± 1.4 (1.02)

**Table 2 molecules-26-07200-t002:** LC-QTOF analysis ethanol extract of *A. santolinifolia*.

No.	Compound Name	Compound Formula	Observed RT (min)	Observed *m*/*z*	Mass Error (mDa)	References
1.	Gramisterol	C_29_H_48_O	25.97	413.3777	−0.1	[23,24]
2.	Momor-cerebroside I	C_48_H_93_NO_10_	25.97	844.6872	0	[25]
3.	Δ7-Stigmasterol	C_29_H_46_O	22.48	411.3622	0	-
4.	Trametenolic acid	C_30_H_48_O_3_	22.49	457.3677	0.1	[26,27,28]
5.	24-Ethyl cholesterol	C_29_H_52_O	25.98	415.3921	−1.3	[29]
6.	Parkeol	C_30_H_50_O	26	427.3925	−0.9	-
7.	Aristolone	C_15_H_22_O	25.29	219.1745	0.2	[30]
8.	Siraitic acid C	C_29_H_44_O_5_	26	441.298	−1.9	-

## Data Availability

The data presented in this study are available on request from the corresponding author. The data are not publicly available due to privacy concerns.

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
