# Peer review of "Artemisia santolinifolia-Mediated Chemosensitization via Activation of Distinct Cell Death Modes and Suppression of STAT3/Survivin-Signaling Pathways in NSCLC"

_molecules, 2021, doi:10.3390/molecules26237200_

Round 1

Reviewer 1 Report

The manuscript by Liu and coworkers focuses on exploring chemosensitization of anticancer drug docetaxel by A. santolinifolia (AS) derived natural products. Docetaxel cotreatment with AS significantly improved its anticancer activity in non-small cell lung cancer A549 and H23 cells. Furthermore, the authors have carried out several biochemical assays to understand the distinct mechanisms of A. santolinifolia in A549 and H23 cells. Overall, this is an interesting study and should be valuable for researchers in the field of natural products. Below are additional comments that need to be addressed before publication. 

  1. Although the authors hypothesized that co-treatment of AS with docetaxel would minimize undesired harmful side effects, it’s not clear how this combination therapy will selectively target cancer cells over normal cells?
  2. In Figure 1D, the authors showed that cytotoxicity (%) of A549 and H23 cells increases with increasing drugs incubation time. However, cytotoxicity (%) decreased with increasing concentrations of AS and DTX in Figures 1A and 1B. The Y-axis label of Figures 1A and 1B is not correct.

  3. In Figure 1C legend, the authors stated that “CI < 1 indicated synergistic effect; CI=1 additivity, while CI > 1 indicated antagonism”.  However, on page 3, line 99, the authors stated that "In general, synergism (CI>1) was observed at high inhibition levels in concurrent treatment with a low dose of DTX, while additivity to antagonism (CI=1, CI<1) was observed at the high dose of DTX.” This statement is no consistent with the Figure 1C legend and needs to be corrected.

  4. The authors have made several errors while referring to the Figure numbers in the result sections. For example, on page 4, "However, the results showed a higher necrotic population, specifically in A549 cell line, when treated with AS alone and in the combined treatment group (P < 0.05, Figure 3A). As shown in Figure 3B, AS alone or merged with DTX induced less necrosis but more apoptotic death in the H23 cell line." Figures number 3A and 3B should be 2A and 2B. Similarly, on page 6, line 170, Figure 2E, F should be Figure 3E, F.

  5. Page 5, “In our previous study, we observed that AS alone could inhibit NSCLC growth through distinct cell death modes depending on the cell line, with concomitant generation of cellular ROS level, induction lipid peroxidation and reduction of glutathione peroxidase 4 (GPX4) expression selectively in A549, while has triggered caspase-dependent apoptosis in H23 cells.” The reference is missing.

  6. The authors stated that “Today, traditional medicine has attracted more attention since it is an excellent source of new drug discovery with minor adverse effects”. However, traditional medicine as a source of new drugs has been known for many years.

  7. Figure 3D, why there is a question mark in the Y-axis label?

  8. In the Figure 3 legend, please provide the details of experiments such as concentration of drugs, incubation time, etc.

  9. The authors identified a list of natural products from ethanol extract of A. santolinifolia. Did the authors confirm their identity by comparing their retention time with authentic samples?

  10. Can the authors speculate the reason for the different mechanisms of Artemisia Santolinifolia in two different cell lines?

  11. There are several grammatical errors throughout the manuscript. Also, some sentences are redundant and as a result, it is difficult to understand the content of the manuscript. Hence, it is important to write clear and precise sentences and carefully proofread the manuscript before resubmission.

Author Response

Dear Reviewer 1,

We sincerely appreciate all valuable comments and suggestions of the Reviewers’ concerning our manuscript entitled “molecules-1464160”. We have carefully considered the statements and tried our best to address every one of them accordingly. The primary corrections in the paper and our responses to all the comments are as follows:

  1. Although the authors hypothesized that co-treatment of AS with docetaxel would minimize undesired harmful side effects, it’s not clear how this combination therapy will selectively target cancer cells over normal cells?

Thanks for your comment. In our preliminary cytotoxic experiment, as a single agent, the cytotoxic dose of AS needs to be as high as 200 ug/mL. Therefore, we believe that AS is a relatively non-toxic herbal extraction. We agree that if there is a cytotoxic profile of AS toward the normal cell would be better. We would consider this aspect in our future works.

  1. In Figure 1D, the authors showed that cytotoxicity (%) of A549 and H23 cells increases with increasing drugs incubation time. However, cytotoxicity (%) decreased with increasing concentrations of AS and DTX in Figures 1A and 1B. The Y-axis label of Figures 1A and 1B is not correct.

Thanks for your kindly notification. The Y-axis label of the manuscript has been changed from “Cytotoxicity (%)” into “Cell viability (%)”.

  1. In Figure 1C legend, the authors stated that “CI < 1 indicated synergistic effect; CI=1 additivity, while CI > 1 indicated antagonism”. However, on page 3, line 99, the authors stated that "In general, synergism (CI>1) was observed at high inhibition levels in concurrent treatment with a low dose of DTX, while additivity to antagonism (CI=1, CI<1) was observed at the high dose of DTX.” This statement is no consistent with the Figure 1C legend and needs to be corrected.

Thanks for the kindly notice. We have revised the symbol in the result section and highlighted it.

  1. The authors have made several errors while referring to the Figure numbers in the result sections. For example, on page 4, "However, the results showed a higher necrotic population, specifically in A549 cell line, when treated with AS alone and in the combined treatment group (P < 0.05, Figure 3A). As shown in Figure 3B, AS alone or merged with DTX induced less necrosis but more apoptotic death in the H23 cell line." Figures number 3A and 3B should be 2A and 2B. Similarly, on page 6, line 170, Figure 2E, F should be Figure 3E, F.

Thanks for the kindly notice. All mistakes were revised according to the comment and highlighted in the revised version.

  1. page 5, “In our previous study, we observed that AS alone could inhibit NSCLC growth through distinct cell death modes depending on the cell line, with concomitant generation of cellular ROS level, induction lipid peroxidation and reduction of glutathione peroxidase 4 (GPX4) expression selectively in A549, while has triggered caspase-dependent apoptosis in H23 cells.” The reference is missing.

Thanks for your comment. Indeed, our previous study, which discovered the possible cytotoxic effect of AS alone, is submitted for journal review. Therefore, it is not proper to put the data in this one. Following your concerns, we have included brief coincident results from our previous experiments below:

As shown in the figure, AS selectively decreased the level of GPX4 in the A549 cell line in a concentration-dependent manner. Conversely, there was nearly no alteration in GPX-4 levels in H23 cells. Further analysis of ROS level by DFCH-DA showed AS triggered significant ROS increase selectively in the A549 cell line but not in H23 cells. Co-treatment of cells with the potent ferroptosis inhibitor (DFO) completely abrogated the AS-induced population of ROS-positive cells without any effect in H23 cells.

  1. The authors stated that “Today, traditional medicine has attracted more attention since it is an excellent source of new drug discovery with minor adverse effects”. However, traditional medicine as a source of new drugs has been known for many years.

We appreciate your comment. The sentence was modified and rephrased to “Traditional medicine has been used since the ancient times….”.

  1. Figure 3D, why is there a question mark in the Y-axis label?

Thanks for your kind notice. It is a software issue. We have corrected the label.

  1. In the Figure 3 legend, please provide the details of experiments such as concentration of drugs, incubation time, etc.

Thank you for your comment. The Figure 3 legend was re-written in accordance and highlighted.

  1. The authors identified a list of natural products from ethanol extract of A. santolinifolia. Did the authors confirm their identity by comparing their retention time with authentic samples?

Thank you for addressing this point. In this study, we aimed to explore the anticancer possibility of A.santolinifolia extract. Accordingly, no extra separation techniques of compounds were used to identify different fractions of AS. However, we realized that the composition of AS might be crucial for the mechanism of action. In this stage, we could only predict the possibility of some active components possessing chemosensitization properties. We applied the small molecule identification based on the suggested information of the UNIFI system (Waters Corp.) in this research. The UNIFI system builds a commercial DATA base with the detailed RT and fragments information if the user follows its specific analysis and acquisition condition. Besides, the user needs to perform detector setup, mass calibration, and internal lock mass profile setup whenever the analysis is performed. During sample analysis, the internal mass calibrator ( Leucine enkephalin in our study) will be monitored every 30-sec simultaneously. Therefore, the system claims that it is not necessary to use individual standards for qualitative detection.

  1. Can the authors speculate the reason for the different mechanisms of Artemisia Santolinifolia in two different cell lines?

As mentioned above, in our previous study, AS selectively induced ROS accumulation and lipid peroxidation only in A549 cells and was prevented when the cells were treated with the iron-chelator DFO. It is similar to the previous research of Shimada and colleagues in which they discovered that ferroptosis inducers were more cell-line selective than other compounds. That might be one prospective explanation relevant to our results that AS selectively induced ferroptosis toward a particular cell line. Also, the findings from the other study conclude that NRF2 protein level balance in A549 and H23 cells impact the overall response to cisplatin treatment, where immunofluorescence images of NRF2 were highly expressed in the nucleus of A549 cells while significantly decreased in the NCI H23 cell. In consist with this report, our study revealed that AS selectively inhibited NRF2 and GPX4 expression in A549. However, slightly increased NRF2 level in H23, giving favor to evading oxidative stress. We predict that overall, the cell initial redox balance, mainly including the critical signaling molecule NRF2 expression, could provide a reasoned explanation of the results.

  1. There are several grammatical errors throughout the manuscript. Also, some sentences are redundant and as a result, it is difficult to understand the content of the manuscript. Hence, it is important to write clear and precise sentences and carefully proofread the manuscript before resubmission.

Thanks for your suggestion. We have searched them through the whole article and improved the wording with the help of a commercial editor company.

Reviewer 2 Report

In the paper “Artemisia Santolinifolia- mediated chemosensitization via activation of distinct cell death modes and suppression of 3 STAT3/Survivin-signaling pathways in NSCLC” explore the potential of a combination therapy of docetaxel (DTX) and Artemisia  Santolinifolia extracts (AS) on for the treatment of Non-small cell lung cancer. The authors observe in  A549 and H20 cell lines a synergistic effect of DTX and AS on cell death. However, the authors describe that A549 cells mainly die via ferroptosis in a GPX4-dependent pathway, while H20 cells die via reduced STAT3 signaling. The paper provides interesting new insides how plant-derived compounds can be used in therapy. Following points should be addressed by the authors in a revised manuscript:

Cell death detection and discrimination: The authors show In Figure 2 a FACS analysis which they use to discriminate between apoptosis and necrosis. This experiment was performed after 48h of treatment. Cell death however is a dynamic process and often ferroptosis can be hard to discriminate from secondary necrosis after apoptosis. Therefore, more time points are needed to exclude differences in the kinetics of cell death.

Caspase 3 / GPX4 Western Blots: Similar to the previous point, multiple time points of caspase cleavage for the combination therapy should be included, to see the kinetic of cell death.

Mass spectrometry: From the provided methods it is unclear, how the authors confirmed the identification of compounds 1-8. Were isotopic standards used as controls? Which fragment ions were detected. The authors should provide chromatograms as supplemental information.

The authors identify multiple compounds in the AS extracts, to which they attach biological function, mainly from literature sources. However, key experiments should be repeated with isolated/purchased compounds.

Minor point: Figure 3D check the y-axis label

Author Response

Dear Reviewer 2,

We sincerely appreciate all valuable comments and suggestions of the Reviewers’ concerning our manuscript entitled “molecules-1464160”. We have carefully considered the statements and tried our best to address every one of them accordingly. The primary corrections in the paper and our responses to all the comments are as follows:

  1. Cell death detection and discrimination: The authors show In Figure 2 a FACS analysis which they use to discriminate between apoptosis and necrosis. This experiment was performed after 48h of treatment. Cell death however is a dynamic process and often ferroptosis can be hard to discriminate from secondary necrosis after apoptosis. Therefore, more time points are needed to exclude differences in the kinetics of cell death.

Thank you for addressing this point. In our preliminary experiment results, we used three different time points to detect the chemosensitization property of AS when combined with DTX. Accordingly, in this study, we choose 48 h, which is the time point of the most significant combination therapeutic effect.  Following the Reviewer's comment, we have included some results from our previous study, which is currently under review in another journal. There were three sample groups, where A549 and H23 cells were treated with AS at 100 and 200 μg/mL concentrations for 24 h. The results were constant with this study data.  7AAD/Annexin V staining revealed AS displayed a dose-dependent increase in cell death but different patterns in two cell lines. AS treatment clustered cells in the upper left quadrant (7-AAD+/Annexin V−) from 3.89% to 25.0% in A549, whereas n H23 cells, AS induced early and late apoptosis from 2.51%, 2.35% to 13.6%, 9.95% at a higher concentration of AS, respectively.

  1. Caspase 3 / GPX4 Western Blots: Similar to the previous point, multiple time points of caspase cleavage for the combination caspase cleavage for the combination therapy should be included, to see the kinetic of cell death.

Thank you for your suggestion. Corresponding to the previous question, we included some results from our previous study, where A549 and H23 cells were treated with AS at 100 and 200 μg/mL concentrations for 24 h. Overall, with our previous data, we could conclude that at these two time points (24 and 48 h), AS-induced alterations that occurred in both cell lines are constant while treated with AS alone or in a combination treatment group.  

  1. Mass spectrometry: From the provided methods it is unclear, how the authors confirmed the identification of compounds 18. Were isotopic standards used as controls? Which fragment ions were detected. The authors should provide chromatograms as supplemental information.

Thanks for your commended. The small molecule identification in this research was based on the suggested information of the UNIFI system (Waters Corp.). The UNIFI system builds a commercial DATA base with the detailed RT and fragments information if the user follows its specific analysis and acquisition condition. Besides, the user needs to perform detector setup, mass calibration, and internal lock mass profile setup whenever the analysis is performed. During sample analysis, the internal mass calibrator ( Leucine enkephalin in our study) will be monitored every 30-sec simultaneously. Therefore, the system claims that it is not necessary to use individual standards for qualitative detection.

  1. The authors identify multiple compounds in the AS extracts, to which they attach biological function, mainly from literature sources. However, key experiments should be repeated with isolated/purchased compounds.

In this study, we aimed to explore the anticancer possibility of A.santolinifolia extract. Accordingly, no extra separation techniques of compounds were used to identify different fractions of AS. However, we realized that the composition of AS might be crucial for the mechanism of action. In this stage, we could only predict the possibility of some active components possessing chemosensitization properties. Highly appreciate your suggestion. We will consider this aspect in future studies.

Minor point: Figure 3D check the y-axis label

Thanks for your kind notice. We have corrected the label in the revised version.

Reviewer 3 Report

  1. The authors need check the Y-axis titles of Figure1 A and B.
  2. “2.2. Enhancement of DXT induced cytotoxicity effect through distinct cell death modalities” should be changed to “2.2. Enhancement of DTX induced cytotoxicity effect through distinct cell death modalities”.
  3. The bar chart does not mark significant differences in Figure 1D.
  4. In Figure 4A, the western blot band of β-Actin in H-23 cell line needs to be replaced.
  5. In my view, to ascertain the possible chemical composition–function relationship, the authors could test whether the eight most abundant compounds mediate chemo sensitization by activating different cell death patterns in non-small cell lung cancer.

Author Response

Dear Reviewer 3,

We sincerely appreciate all valuable comments and suggestions of the Reviewers’ concerning our manuscript entitled “molecules-1464160”. We have carefully considered the statements and tried our best to address every one of them accordingly. The primary corrections in the paper and our responses to all the comments are as follows:

  1. The authors need check the Y-axis titles of Figure1 A and B.

Thanks for your kindly notification. The Y-axis label of the manuscript has been changed from “Cytotoxicity (%)” into “Cell viability (%)”.

  1. “2.2. Enhancement of DXT induced cytotoxicity effect through distinct cell death modalities” should be changed to “2.2. Enhancement of DTX induced cytotoxicity effect through distinct cell death modalities”.

Thank you for your notice. We have corrected the word spelling “DXT” to “DTX”.

  1. The bar chart does not mark significant differences in Figure 1D.

Thanks for your comment. Indeed, treated as a single agent, DTX sub-optimal dose partially affected the H23 cell viability but almost didn’t affect the cell proliferation in the A549 cell in Figure 1D. It may be the reason that in comparison of DTX group with combination group in H23 cells bar chart, we couldn’t see the statistical significance.

  1. In Figure 4A, the western blot band of β-Actin in H-23 cell line needs to be replaced.

We appreciate your comment. As the Reviewer suggested, we replaced the β-actin band in the H23 cell.

  1. In my view, to ascertain the possible chemical composition–function relationship, the authors could test whether the eight most abundant compounds mediate chemo sensitization by activating different cell death patterns in non-small cell lung cancer.

Highly appreciate your suggestion. We will conduct the chemosensitization experiment with pure chemicals in future studies.

Round 2

Reviewer 1 Report

The authors have carefully reviewed my comments and have considerably responded in a point-by-point manner. After reexploring the revised manuscript, I think it is quite acceptable and have no further suggestions.

Reviewer 2 Report

The authors reviewed and answered my questions and edited the manuscript accordingly. The paper can be published in its present form

Reviewer 3 Report

Accept